# Self-Supervised Generalisation with Meta Auxiliary Learning

## Abstract

Auxiliary learning has been shown to improve the generalisation performance of a principal task. But typically, this requires manually-defined auxiliary tasks based on domain knowledge. In this paper, we consider that it may be possible to automatically learn these auxiliary tasks to best suit the principal task, towards optimum auxiliary tasks without any human knowledge. We propose a novel method, Meta Auxiliary Learning (MAXL), which we design for the task of image classification, where the auxiliary task is hierarchical sub-class image classification. The role of the meta learner is to determine sub-class target labels to train a multi-task evaluator, such that these labels improve the generalisation performance on the principal task. Experiments on three different CIFAR datasets show that MAXL outperforms baseline auxiliary learning methods, and is competitive even with a method which uses human-defined sub-class hierarchies. MAXL is self-supervised and general, and therefore offers a promising new direction towards automated generalisation.

## 1 Introduction

Auxiliary learning is a method to improve the generalisation of a task. It works by training on additional auxiliary tasks simultaneously with the principal task. Extra data may be available for those auxiliary tasks, but not the principal task. If the auxiliary tasks and the principal task share some common reasoning, then the prediction model is encouraged to learn additional relevant features which otherwise would not be learned from single-task learning. The broader support of these features then assists with generalisation of the principal task.

We now rethink this generalisation by considering that not all auxiliary tasks are created equal. In supervised auxiliary learning (Liebel & Körner, 2018; Toshniwal et al., 2017), auxiliary tasks can be carefully chosen to complement the principal task, but at the expense of a dependency on labelled data. Unsupervised auxiliary learning (Flynn et al., 2016; Zhou et al., 2017; Zhang et al., 2018; Jaderberg et al., 2017) alleviates this, but at the expense of a limited set of auxiliary tasks which may not be well aligned with the principal task. By combining the merits of both supervised and unsupervised auxiliary learning, the ideal auxiliary learning framework is one with the flexibility to automatically determine the optimum auxiliary tasks, but without the requirement of any manually-labelled data.

In this paper, we propose to achieve such a framework with a simple and general meta-learning algorithm which we call Meta AuXiliary Learning (MAXL). Given a principal task, the goal of MAXL is to discover the auxiliary tasks which, when trained alongside the principal task, give the greatest generalisation performance of the principal task on a meta dataset. In our work, we focus on the problem of image classification, where an auxiliary task is required to assign a sub-class label to an image. As such, data is classified both at a coarse level as the principal task, and at a fine level as the auxiliary task. The meta learner's role is then to determine the target labels for this sub-class labelling, in such a way that the learned features induced by learning these additional, more complex auxiliary tasks generate the best generalisation performance for the principal task.

As well as our method being able to automatically learn the optimum auxiliary tasks, we achieve this in an unsupervised manner, giving potential to scale well beyond any datasets without manually-labelled auxiliary tasks, such a class hierarchy as in our experiments. And even when such a hierarchy is available, in our experiments we show that MAXL is at least as competitive despite

this hierarchy being learned in an unsupervised manner. In our experiments, we define the auxiliary tasks as sub-class labelling with MAXL learning to generate target sub-class labels, but MAXL is general and in future work this could be relaxed to actually learn the auxiliary tasks themselves. The ability to learn these tasks in a purely unsupervised and scalable manner opens up an exciting new way of thinking about how we can achieve generalisation in an automated manner.

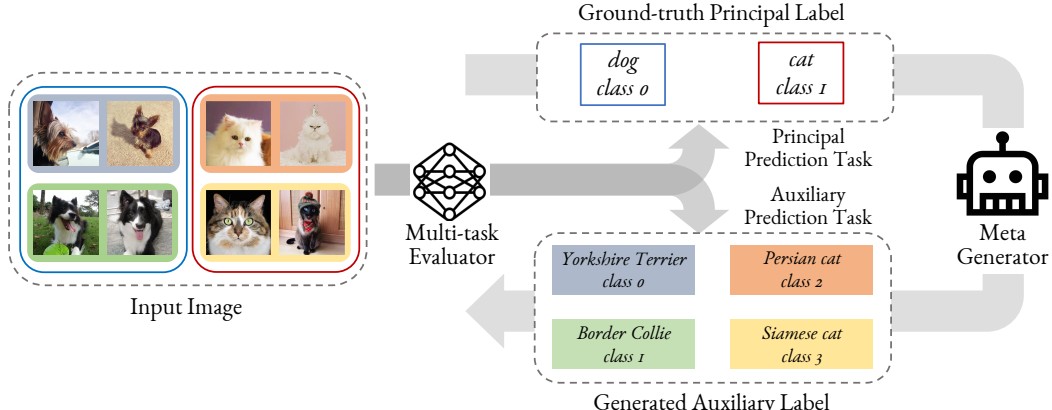

Figure 1: Illustration of our proposed MAXL framework. The Multi-task evaluator takes an input image and is trained to predict both the principal class (e.g. Dog), and the auxiliary class (e.g. Border Collie). The principal class has a ground-truth label, but the label for the auxiliary class is determined by the meta generator. The meta generator is trained by outputting auxiliary class labels which, when used to train the multi-task evaluator, improve its prediction performance on the principal task.

## 2 RELATED WORK

This work brings ideas together from a number of related areas of machine learning.

**Multi-task & Transfer Learning**   The aim of multi-task learning (MTL) is to achieve shared representations by simultaneously training a set of related learning tasks. In this case, the learned knowledge used to share across domains is encoded into the feature representations, to improve performance of each individual task, since knowledge distilled from related tasks are interdependent. The success of deep neural networks has led to some recent methods advancing the multi-task architecture design, such as applying a linear combination of task-specific features (Misra et al., 2016; Doersch & Zisserman, 2017; Kokkinos, 2017). Liu et al. (2018) applied soft-attention modules as feature selectors, allowing learning of both task-shared and task-specific features in a self-supervised, end-to-end manner. Transfer learning is another common approach to improve generalisation, by incorporating knowledge learned from one or more related domains. Pre-training a model with a large-scale dataset such as ImageNet (Deng et al., 2009) has become standard practise in many vision-based applications. The transferability of different convolutional layers in CNNs has also been investigated in Yosinski et al. (2014).

**Auxiliary Learning**   Whilst in multi-task learning the goal is high test accuracy across all tasks, auxiliary learning differs in that high test accuracy is only required for a single principal task, and the role of the auxiliary tasks is to assist in generalisation of this principal task. Toshniwal et al. (2017) applied auxiliary supervision with phoneme recognition at intermediate low-level representations of deep networks to improve the performance of conversational speech recognition. Liebel & Körner (2018) chose auxiliary tasks which can be obtained with low effort, such as global descriptions of a scene, to boost the performance for single scene depth estimation and semantic segmentation. By carefully choosing a pair of learning tasks, we may also perform auxiliary learning without ground truth labels, in an unsupervised manner. Jaderberg et al. (2017) introduced a method for improving the learning agents in Atari games, by building unsupervised auxiliary tasks to predict the onset of immediate rewards from a short historical context. Flynn et al. (2016); Zhou et al. (2017) proposed

image synthesis networks to perform unsupervised monocular depth estimation by predicting the relative pose of multiple cameras. Different from these works which require prior knowledge to manually define suitable auxiliary tasks, our proposed method requires no additional task knowledge, since our meta learner generates useful auxiliary knowledge in a purely unsupervised fashion. The most similar work to ours is Zhang et al. (2018), in which meta learning was used in auxiliary data selection. However, this still requires manually-labelled data from which these selections are made, whilst our method is able to generate auxiliary data from scratch.

**Meta Learning**    Meta learning (or learning to learn) aims to design a higher-level learning system which itself is trained using the experiences of a lower-level learning system, in an attempt to improve this lower-level system. Early works in meta learning explored automatically learning update rules for neural models (Bengio et al., 1990; 1992; Schmidhuber, 1992). Recent approaches have focused on learning optimisers for deep networks based on LSTMs (Ravi & Larochelle, 2016) or synthetic gradients (Andrychowicz et al., 2016; Jaderberg et al., 2016). Meta learning has also been studied for finding optimal hyper-parameters (Li et al., 2017) and a good initialisation for few-shot learning (Finn et al., 2017). (Santoro et al., 2016) also investigated few shot learning via an external memory module. Vinyals et al. (2016); Snell et al. (2017) realised few shot learning in the instance space via a differentiable nearest-neighbour approach. Our method also performs in the instance space, but induces auxiliary knowledge as an implicit regularisation to improve generalisation of the principal task.

## 3    META AUXILIARY LEARNING

In this section, we introduce our method for automatically generating optimum auxiliary tasks, which we call Meta AuXiliary Learning (MAXL).

### 3.1    PROBLEM SETUP

The goal of meta auxiliary learning is to train a meta generator that can generate higher complexity auxiliary tasks, to improve performance of the principal task. To accomplish this, we use two networks: a *multi-task evaluator* which trains on the principal and auxiliary tasks, and evaluates the performance of the auxiliary tasks on a meta set, and a *meta generator* which generates these auxiliary tasks. For simplicity, we consider image classification tasks in this section, where the auxiliary task is sub-class labelling, and the meta generator determines target sub-class labels, but the approach can be considered general for any type of task.

We denote the multi-task evaluator as a function $f_{\theta_1}(x)$ that takes an input $x$ with network parameters $\theta_1$, and the meta generator as a function $g_{\theta_2}(x)$ that takes the same input $x$ with network parameters $\theta_2$. For a dataset with input $x$ and ground-truth label $y$ for the principal task, we split into three subsets: training $(x_{\text{train}}, y_{\text{train}})$, meta-training $(x_{\text{meta}}, y_{\text{meta}})$, and test $(x_{\text{test}}, y_{\text{test}})$. Training data is used for updating $\theta_1$, meta-training data is used for updating the $\theta_2$, and test data is used for overall evaluation.

In the multi-task evaluator, we apply a hard parameter sharing approach (Ruder, 2017) in which we predict the principal and auxiliary tasks using the shared set of features $\theta_1$ in the multi-task network. At the end of the last feature layer $f_{\theta_1}(x)$, we then apply further task-specific layers to output the corresponding prediction for each task. We denote the predicted principal labels by $f_{\theta_1}^{\text{pri}}(x)$ and predicted auxiliary labels by $f_{\theta_1}^{\text{aux}}(x)$.

In the meta generator, we pre-define a hierarchical structure $\psi$ which determines the number of sub-classes for each class in the principal task. At the end of the last feature layer $g_{\theta_2}(x)$, this hierarchy, together with the ground-truth label $y$ for the principal task, are used to generate the target auxiliary labels, denoted by $g_{\theta_2}^{\text{gen}}(x, y, \psi)$. We allow for soft assignment labelling rather than enforcing one-hot encoding, which enables greater flexibility to learn optimum auxiliary tasks. The meta generator uses a masked SoftMax to ensure that each output node represents a sub-class label for only one class in the principal task, as described further in Section 3.3. The visualisation of the our proposed MAXL approach is shown in Figure 2.

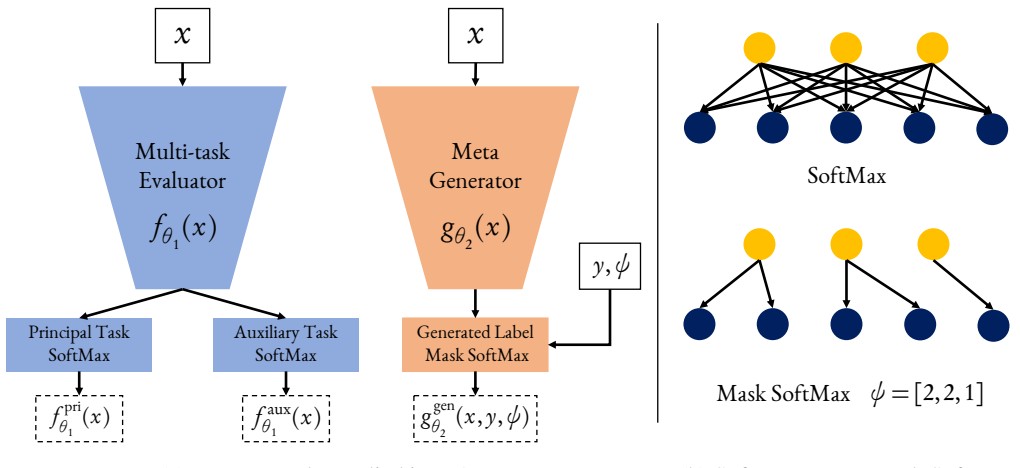

(a) Two networks applied in MAXL

(b) SoftMax versus Mask SoftMax

Figure 2: (a) Illustration of the two networks which make up our meta auxiliary learning algorithm. (b) Illustration of vanilla SoftMax and Mask SoftMax with 3 principal classes. Vanilla SoftMax outputs over all 5 auxiliary classes, where as Mask Softmax outputs over a hierarchical structure $\psi = [2, 2, 1]$ to constrain the prediction space.

## 3.2 MODEL OBJECTIVES

The multi-task evaluator is trained in a tightly-coupled manner with the meta generator: the meta generator determines target labels for the multi-task evaluator, which in turn determines the suitability of those labels.

Given target labels as determined by the meta generator, the multi-task evaluator is trained to predict these labels, alongside the ground-truth labels for the principal task. For both the principal and auxiliary classification tasks, we apply focal loss (Lin et al., 2017) with a focusing parameter $\gamma = 2$, defined as:

$$\mathcal{L}(\hat{y}, y) = -y(1 - \hat{y})^{\gamma} \log(\hat{y}), \tag{1}$$

where $\hat{y}$ is the predicted label and $y$ is the ground-truth label. The focal loss helps to focus on the incorrectly predicted labels, which we found improved performance during our experimental evaluation compared with the regular cross-entropy log loss.

To update parameters $\theta_1$ in the multi-task evaluator, we define the multi-task objective as follows:

$$\arg\min_{\theta_1} \left( \mathcal{L}(f_{\theta_1}^{\mathrm{pri}}(x_{\mathrm{train}}^{(i)}), y_{\mathrm{train}}^{(i)}) + \mathcal{L}(f_{\theta_1}^{\mathrm{aux}}(x_{\mathrm{train}}^{(i)}), g_{\theta_2}^{\mathrm{gen}}(x_{\mathrm{train}}^{(i)}, y_{\mathrm{train}}^{(i)}, \psi)) \right) , \tag{2}$$

where $(i)$ represents the $i^{th}$ batch from the training data.

The meta generator is then trained by encouraging target labels for the auxiliary task to be chosen such that, if the multi-task evaluator were to be trained on these labels, the performance on the principal task would be maximised. This requires evaluation on a separate dataset, the meta-training set, to train the meta generator, to ensure that the target auxiliary labels encourage generalisation beyond the data supplied to the multi-task evaluator.

To update parameters $\theta_2$ in the meta generator, we define the meta objective as follows:

$$\arg\min_{\theta_2} \mathcal{L}(f_{\theta_1^+}^{\mathrm{pri}}(x_{\mathrm{meta}}^{(i)}), y_{\mathrm{meta}}^{(i)}) . \tag{3}$$

Here $\theta_1^+$ represents the weights of the multi-task network were it to be trained, with one gradient update, using auxiliary labels $y_{\mathrm{meta}}$:

$$\theta_1^+ = \theta_1 - \alpha \nabla_{\theta_1} \left( \mathcal{L}(f_{\theta_1}^{\mathrm{pri}}(x_{\mathrm{meta}}^{(i)}), y_{\mathrm{meta}}^{(i)}) + \mathcal{L}(f_{\theta_1}^{\mathrm{aux}}(x_{\mathrm{meta}}^{(i)}), g_{\theta_2}^{\mathrm{gen}}(x_{\mathrm{meta}}^{(i)}, y_{\mathrm{meta}}^{(i)}, \psi)) \right), \tag{4}$$

where $\alpha$ is the learning rate.

The trick in this meta objective is that we perform the derivative over a derivative (a Hessian matrix) to update $\theta_2$, by using a retained computational graph of $\theta_1^+$ in order to compute derivatives with respect to $\theta_2$. This second derivative trick in meta learning was also proposed in Finn et al. (2017) and Zhang et al. (2018).

However, we found that the generated auxiliary labels can easily collapse (i.e. degenerate by simply learning a similar level of complexity as the principal task), which leaves parameters $\theta_2$ in a local minimum without producing any extra useful knowledge. Thus, to encourage the network to learn more complex and informative auxiliary tasks, we further apply an entropy loss $\mathcal{H}(g_{\theta_2}(x_{\text{meta}}^{(i)}, y_{\text{meta}}^{(i)}, \psi))$ as a regularisation term in the meta objective. A detailed explanation of the entropy loss and the collapsing label problem will be given in Section 3.4.

Finally, the entire MAXL algorithm is defined as follows:

---

**Algorithm 1:** The MAXL algorithm

---

**Dataset:** $D = \left\{ (x_{\text{train}}, y_{\text{train}}), (x_{\text{meta}}, y_{\text{meta}}) \right\}$
**Initialise:** Network parameters: $\theta_1, \theta_2$; Hierarchical structure: $\psi$
**Initialise:** Hyper-parameter (learning rate): $\alpha, \beta$; Hyper-parameter (task weighting): $\lambda$
**for** *each training iteration $i$* **do**

> *# fetch one batch of training and meta data*
> $\left\{ (x_{\text{train}}^{(i)}, y_{\text{train}}^{(i)}), (x_{\text{meta}}^{(i)}, y_{\text{meta}}^{(i)}) \right\} \in \left\{ (x_{\text{train}}, y_{\text{train}}), (x_{\text{meta}}, y_{\text{meta}}) \right\}$
> *# training step*
> Update: $\theta_1 \leftarrow \theta_1 - \alpha \nabla_{\theta_1} \left( \mathcal{L}(f_{\theta_1}^{\text{pri}}(x_{\text{train}}^{(i)}), y_{\text{train}}^{(i)}) + \mathcal{L}(f_{\theta_1}^{\text{aux}}(x_{\text{train}}^{(i)}), g_{\theta_2}(x_{\text{train}}^{(i)}, y_{\text{train}}^{(i)}, \psi)) \right)$
> *# meta-training step*
> Compute: $\theta_1^+ = \theta_1 - \alpha \nabla_{\theta_1} \left( \mathcal{L}(f_{\theta_1}^{\text{pri}}(x_{\text{meta}}^{(i)}), y_{\text{meta}}^{(i)}) + \mathcal{L}(f_{\theta_1}^{\text{aux}}(x_{\text{meta}}^{(i)}), g_{\theta_2}(x_{\text{meta}}^{(i)}, y_{\text{meta}}^{(i)}, \psi)) \right)$
> Update: $\theta_2 \leftarrow \theta_2 - \beta \nabla_{\theta_2} \left( \mathcal{L}(f_{\theta_1^+}^{\text{pri}}(x_{\text{meta}}^{(i)}), y_{\text{meta}}^{(i)}) + \lambda \mathcal{H}(g_{\theta_2}^{\text{gen}}(x_{\text{meta}}^{(i)}, y_{\text{meta}}^{(i)}, \psi)) \right)$

**end**

---

### 3.3 MASK SOFTMAX FOR HIERARCHICAL PREDICTIONS

In the prediction layer of the meta generator, we designed a modified SoftMax function to predict target auxiliary labels which conform to a pre-defined hierarchy $\psi$. As shown in Figure 2 (upper right), the original softmax function does not constrain sub-class labelling to lie within this hierarchy. Our mask SoftMax structure resolves this issue by applying a binary mask to the original SoftMax function.

The overall hierarchical structure $\psi$ determines the number of sub-classes $\psi[i]$ in each principal class $i$. As such, the total prediction space for auxiliary labels is $\sum_i \psi[i]$. This hierarchy, together with the ground-truth principal class label $y$ of the current image, creates the mask with a binarise function $M = \mathcal{B}(y, \psi)$. Using the principal ground-truth label $y$, the corresponding range of sub-classes $\psi[y]$ is selected, and a binary mask $M$ is created with size $\sum_i \psi[i]$ with a multi one-hot encoding $\mathbb{1}_{\sum_{i<y} \psi[i]:\sum_{i<y+1} \psi[i]}$ ($\mathbb{1}_{a:b}$ is denoted as a multi one-hot encoding in which indexes from $a$ to $b$ are encoded as 1).

Using the example in Figure 2, consider the principal task to have 3 classes with ground truth labels $y = 0, 1, 2$, and hierarchical structure $\psi = [2, 2, 1]$. In this case, the auxiliary prediction space is equal to 5 and the corresponding binary masks are $M = [1, 1, 0, 0, 0], [0, 0, 1, 1, 0], [0, 0, 0, 0, 1]$ respectively.

Finally, we apply binary mask $M$ with an element-wise multiplication on the original SoftMax function for the final auxiliary task predictions:

SoftMax: $\quad p(\hat{y}_i) = \dfrac{\exp \hat{y}_i}{\sum_i \exp \hat{y}_i}, \qquad$ Mask SoftMax: $\quad p(\hat{y}_i) = \dfrac{\exp M \odot \hat{y}_i}{\sum_i \exp M \odot \hat{y}_i}, \quad M = \mathcal{B}(y, \psi)\,,$

where $p(\hat{y}_i)$ represents the probability of the predicted principal label $\hat{y}$ over class $i$, and $\odot$ represents element-wise multiplication.

### 3.4 THE COLLAPSING CLASS PROBLEM

As previously discussed, we predict each auxiliary label within a hierarchical structure $\psi$. However, the number of sub-classes defined in $\psi[i]$ is the maximum auxiliary label prediction space, with no guarantee that all $\psi[i]$ classes will be predicted. This may result in some auxiliary labels defined in $\psi[i]$ being overlooked, with the output of the meta generator collapsing into a smaller sub-class space. In experiments, we found that this phenomenon is particularly apparent when we either have a large learning rate for training the meta generator, or a large sub-class prediction space $\psi$.

To avoid the collapsing class problem, we introduced an additional regularisation loss, which we call the entropy loss $\mathcal{H}(\hat{y}^{(i)})$. This encourages the meta generator to utilise the full prediction space, by encouraging a large prediction entropy across this space.

Assuming we have a well-balanced dataset, the entropy loss calculates the KL divergence between the predicted auxiliary label space $\hat{y}^{(i)}$, and a uniform distribution $\mathcal{U}$ for each $i^{th}$ batch. This is equivalent to calculating the entropy of the predicted label space, and is defined as:

$$\mathcal{H}(\hat{y}^{(i)}) = \sum_{k=1}^{K} \overline{y_k} \log \overline{y_k}, \quad \overline{y_k} = \frac{1}{N} \sum_{i=1}^{N} \hat{y}^{(i)}. \tag{5}$$

where $K$ is the number of auxiliary labels and $N$ is the training batch size.

The entropy loss is essential to achieve human-level performance, as shown in our experiments. The higher entropy in the auxiliary target labels results in a more complex auxiliary task. This avoids local minima during training, such as assigning a single label to all examples of a principal class.

## 4 EXPERIMENTS

In this section, we present experimental results to evaluate MAXL with respect to several baselines and datasets on image classification tasks.

### 4.1 EXPERIMENTAL SETUP

**Datasets** We evaluated on three different datasets: CIFAR100, CIFAR10, and CIFAR10.1v6 (Recht et al., 2018). CIFAR100 consists of 100 principal classes, whilst CIFAR10 and CIFAR10.1v6 consist of 10 principal classes and have the same training dataset as each other, but two different test datasets. To assess the generalisation across different task complexities, we tested a range of different combinations in the numbers of principal and auxiliary classes. For CIFAR100, we expanded the dataset's provided 2-level hierarchy (20 and 100 classes) into a 4-level hierarchy (additional 3 and 10 classes), by manually assigning examples for these new hierarchy levels (see Appendix A). Based on the new hierarchy, we then tested on all 6 possible combinations of principal and auxiliary class numbers. Note that for MAXL, the hierarchy was used only to define the structure of $\psi$ and the principal task labels, to ensure a fair comparison with a method using human-defined auxiliary tasks, but the auxiliary task labelling within that structure was learned by MAXL itself. CIFAR10 and CIFAR10.1v6 do not have an associated manually-defined hierarchy, and so we defined a range of hierarchical structures $\psi[i] = 2, 5, 10, 20, 50, 100, \forall i$.

**Baselines** We compared MAXL to a number of baselines. *Single Task* trains only with the principal class label. *Random Assignment* trains with auxiliary classes, and randomly assigns the auxiliary class labels. *Prototypical Net* is a clustering method based on (Snell et al., 2017), where prototypes for auxiliary classes are defined by embedding examples from meta-training data, which has human-defined auxiliary classes, using a pre-trained ImageNet network. Unsupervised, differentiable, nearest-neighbour clustering is then used to produce the final auxiliary class labelling for the remaining training data. The key difference to MAXL is that, whilst both methods are unsupervised, the auxiliary class labelling with MAXL actually evaluates the generalisation performance of this labelling on the principal task, whilst the Prototypical Net method does not. Finally, *Human* trains with auxiliary classes, using the human-defined hierarchy. Note that due to the need for a manually-defined hierarchy, Prototypical Net and Human were only evaluated on CIFAR100. For all baselines, we use the same network architecture and training procedure as MAXL's multi-task

evaluator. For the meta-training for MAXL and Prototypical Net, we split each training dataset and used 10% for meta-training the auxiliary labelling, and 90% for training the multi-task evaluator. For all other baselines, we used the full training set for training the multi-task evaluator.

**Training** For both the multi-task evaluator and the meta generator use VGG-16 as its core (Simonyan & Zisserman, 2014), together with batch normalisation. For all experiments, we used a learning rate of 0.01 for the multi-task evaluator. For MAXL's meta generator, we found that a smaller learning rate of $10^{-5}$ was necessary to help prevent the class collapsing problem. For all training, we drop the learning rate by half after every 50 epochs, and train for a total of 200 epochs, using vanilla stochastic gradient descent. For the meta generator, we apply an $L_1$ norm weight decay of $5 \cdot 10^{-4}$ on the meta generator, with no regularisation on the multi-task evaluator. We chose the weighting of the entropy regularisation loss term to be 0.2 based on empirical performance.

## 4.2 TEST PERFORMANCE

We now evaluate the performance of MAXL compared to these baselines, on all three datasets. Results for CIFAR100 are presented in Figure 3, and results for CIFAR10 and CIFAR10.1v6 are presented in Appendix B.

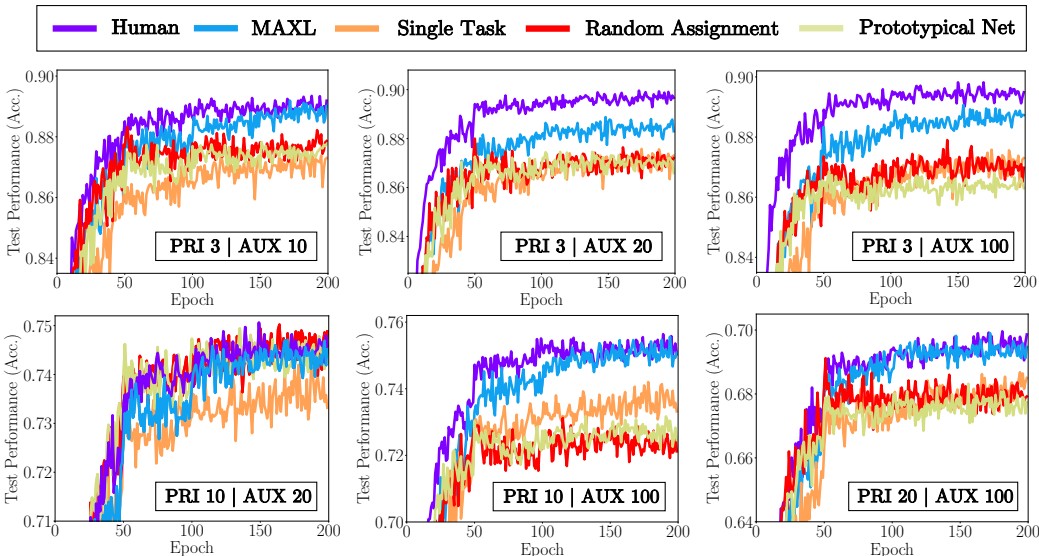

Figure 3: Learning curves for the CIFAR100 test dataset, comparing MAXL with baseline methods. We provide results in all 6 different combinations of principal and auxiliary class numbers.

For CIFAR100, we observe that MAXL performs similarly to when human knowledge is used in 4 out of the 6 hierarchical structures, and performs worse in 2 out of the 6. For all other baselines, MAXL performs at least as well, and in the majority of cases outperforms other baselines by a significant margin. We therefore see that MAXL is able to learn auxiliary tasks effectively by tightly coupling the auxiliary task generation and the principal task training, in a superior manner than when these auxiliary tasks are assigned independently, such as with random assignment or using prototypical net. With performance of MAXL approaching that of a system using a human-defined auxiliary tasks, we see strong evidence that MAXL is able to learn to generalise effectively in an unsupervised manner.

## 4.3 EFFECT OF AUXILIARY TASK COMPLEXITY

We now evaluate how the complexity of the auxiliary tasks affects the performance of the principal task. In Figure 4 (a), we present results from CIFAR10 and CIFAR10v1.6 showing the performance increase over single-task learning, when there are 10 principal classes, but a range of auxiliary class numbers ($\psi[i] = 2, 5, 10, 20, 50, 100, \forall i$). For each data point, the performance is calculated by

averaging the test accuracy from the last 5 epochs, after a total of 200 epochs. Experiments were performed both with and without the entropy loss term to show the benefit of this regularisation.

We observe an interesting trend in which test performance rises as the number of auxiliary classes increases, but then begins to fall. This suggests that for a given complexity of principal task, there is an optimum complexity in the auxiliary tasks. One explanation for this may be that as the auxiliary tasks increase in complexity, the learned features favour learning these auxiliary tasks rather than the principal task, encouraging further generalisation beyond the features learned only for the principal task. But if the auxiliary task is too complex, then these features begin to overfit and lose the overlap between the reasoning required for the principal and auxiliary tasks, begins to decrease.

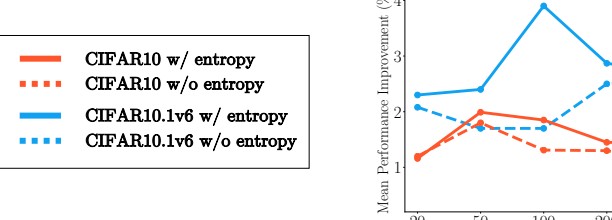

Figure 4: Performance improvement in percentages when training with MAXL compared with single-task learning, with 10 principal classes and a range of auxiliary classes.

## 4.4 VISUALISATIONS OF GENERATED KNOWLEDGE

In Figure 5, we visualise 2D embeddings of examples from the CIFAR100 test dataset, on two different task complexities. This was computed using t-SNE (Maaten & Hinton, 2008) on the final feature layer of the multi-task evaluator, and compared across three methods: our MAXL method, our baseline using human-defined hierarchy, and our baseline using single-task learning.

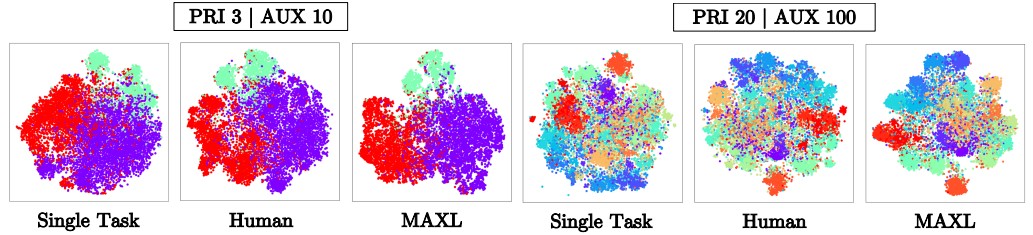

Figure 5: t-SNE visualisation of the learned final layer of the multi-task evaluator network, trained with two combinations of principal and auxiliary class numbers from CIFAR100. Colours represent the principal classes.

This visualisation shows the separability of principal classes after being trained with the multi-task evaluator. We see that both MAXL and Human show better separation of the principal classes than with Single-Task, owing to the generalisation effect of the auxiliary task learning. The distinction between the separability of the MAXL and Human visualisations is not as clear, despite their very similar performance for these two task complexities in Figure 3. But given that MAXL uses the same hierarchical structure as Human, we see from the visualisation that these two methods are clearly learning different representations.

We also show examples of images assigned to the same auxiliary class through MAXL's multi-task evaluator. Figure 6 shows example images with the highest prediction probabilities for three random auxiliary classes from CIFAR100, using the combination of 20 principal classes and 5 auxiliary classes per principal class, which showed the best performance of MAXL in Figure 3. In addition, we also applied MAXL to MNIST, in which 3 auxiliary classes were used for each of the 10 principal classes.

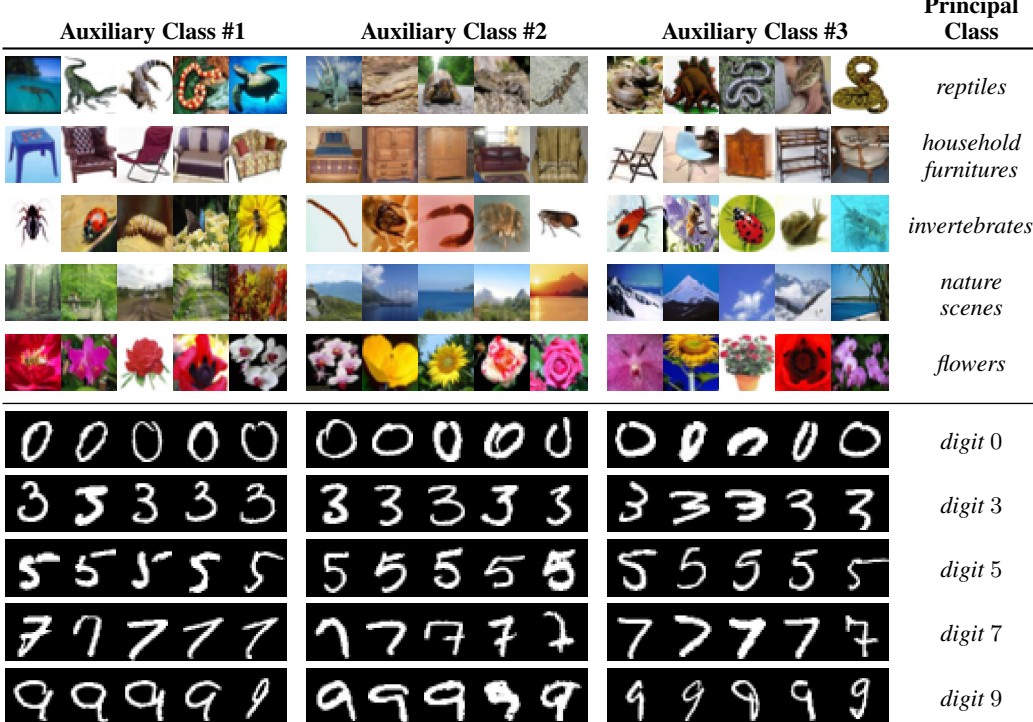

|  | Auxiliary Class #1 | Auxiliary Class #2 | Auxiliary Class #3 | Principal Class |

Figure 6: Visualisation of 5 test examples with the highest prediction probability, for each of 3 randomly selected auxiliary classes, for a number of different principal classes. We present the visualisation for CIFAR100 (top) when trained with 20 principal classes and 5 auxiliary classes per principal class, and for MNIST (bottom) when trained with 10 principal classes and 3 auxiliary classes per principal class.

To our initial surprise, the generated auxiliary labels visualised in both datasets show no clear human-understandable knowledge. In particular, there are no obvious similarities within each auxiliary class whether in terms of shape, colour, style, structure or semantic meaning. However, this makes more sense when we re-consider the task of the meta generator, which is to assign auxiliary labels which assist the principal task. Rather than grouping images in terms of semantic or visual similarity, the meta generator would therefore be more effective it it were to group images in terms of a shared aspect of reasoning which the multi-task evaluator is currently facing difficulty on. If the multi-task evaluator is then able to improve its ability to determine the auxiliary class of an image in such a cluster, then the learned features will help in overcoming this challenging aspect of reasoning. It therefore makes sense that the examples within an auxiliary class do not share semantic or visual similarity, but instead share a more complex underlying property.

Further, we discovered that the generated auxiliary knowledge is not deterministic, since the top predicted candidates are different when we re-train the network from scratch. We therefore speculate that using a human-defined hierarchy is just one out of a potentially infinite number of local optima, and on each run of training the meta generator produces another of these local optimums.

## 5    CONCLUSION & FUTURE WORK

In this paper, we have presented and evaluated Meta AuXiliary Learning (MAXL). MAXL learns to generate optimum auxiliary tasks which, when trained alongside a principal task in a multi-task setup, maximise the generalisation of the principal task across a validation dataset. Rather than employing domain knowledge and human-defined auxiliary tasks as is typically required, MAXL is self-supervised and, combined with its general nature, has the potential to automate the process of generalisation to new levels.

Our evaluations on three image datasets have shown the performance of MAXL in an image classification setup, where the auxiliary task is to predict sub-class, hierarchical labels for an image. We have shown that MAXL significantly outperforms other auxiliary learning baselines, and even when human-defined knowledge is used to manually construct the auxiliary tasks, MAXL performs similarly in the majority of experiments.

Despite this impressive performance from a self-supervised method, questioning why auxiliary tasks generated by MAXL do not outperform those constructed by a human opens exciting future research in this direction. Perhaps, human-defined auxiliary tasks are optimal themselves and cannot be surpassed. However, we believe this not to be the case since such tasks are typically chosen due to the availability of labelled data for these tasks, and not necessarily their optimality when combined with the principal task. Alternatively, perhaps the power of the human knowledge is not from the domain specific labels, but from higher-level reasoning about how auxiliary tasks should be structured. In our experiments, training MAXL using the same structure as a human-defined hierarchy, but learning its own auxiliary labels, typically led to similar performance as when the human-defined labels were used.

The general nature of MAXL also opens up questions about how self-supervised auxiliary learning may be used to learn generic auxiliary tasks beyond sub-class labelling. During our experiments, we also ran preliminary experiments on predicting arbitrary vectors as the auxiliary task, but results so far have been inconclusive. However, the ability of MAXL to potentially learn flexible auxiliary tasks which can automatically be tuned for the principal task now offers an exciting direction towards automated generalisation across a wide range of more complex tasks.

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

# A  4-LEVEL CIFAR100 DATASET

Table 1: Building a 4-level hierarchy for image classification task based on CIFAR100 dataset. Originally, a 20-class and 100-class heirarchiy was provided, and we manually introduced a 3-class and 10 class layer.

| 3 Class | 10 Class | 20 Class | 100 Class |
|---|---|---|---|
| animals | large animals | reptiles | crocodile, dinosaur, lizard, snake, turtle |
| | | large carnivores | bear, leopard, lion, tiger, wolf |
| | | large omnivores and herbivores | camel, cattle, chimpanzee, elephant, kangaroo |
| | medium animals | aquatic mammals | beaver, dolphin, otter, seal, whale |
| | | medium-sized mammals | fox, porcupine, possum, raccoon, skunk |
| | small animals | small mammals | hamster, mouse, rabbit, shrew, squirrel |
| | | fish | aquarium fish, flatfish, ray, shark, trout |
| | invertebrates | insects | bee, beetle, butterfly, caterpillar, cockroach |
| | | non-insect invertebrates | crab, lobster, snail, spider, worm |
| | people | people | baby, boy, girl, man, woman |
| vegetations | vegetations | flowers | orchids, poppies, roses, sunflowers, tulips |
| | | fruit and vegetables | apples, mushrooms, oranges, pears, peppers |
| | | trees | maple, oak, palm, pine, willow |
| objects and scenes | household objects | food containers | bottles, bowls, cans, cups, plates |
| | | household electrical devices | clock, keyboard, lamp, telephone, television |
| | | household furniture | bed, chair, couch, table, wardrobe |
| | construction | large man-made outdoor things | bridge, castle, house, road, skyscraper |
| | natural scenes | large natural outdoor scenes | cloud, forest, mountain, plain, sea |
| | vehicles | vehicles 1 | bicycle, bus, motorcycle, pickup truck, train |
| | | vehicles 2 | lawn-mower, rocket, streetcar, tank, tractor |

# B LEARNING CURVES FOR CIFAR10/10.1V6

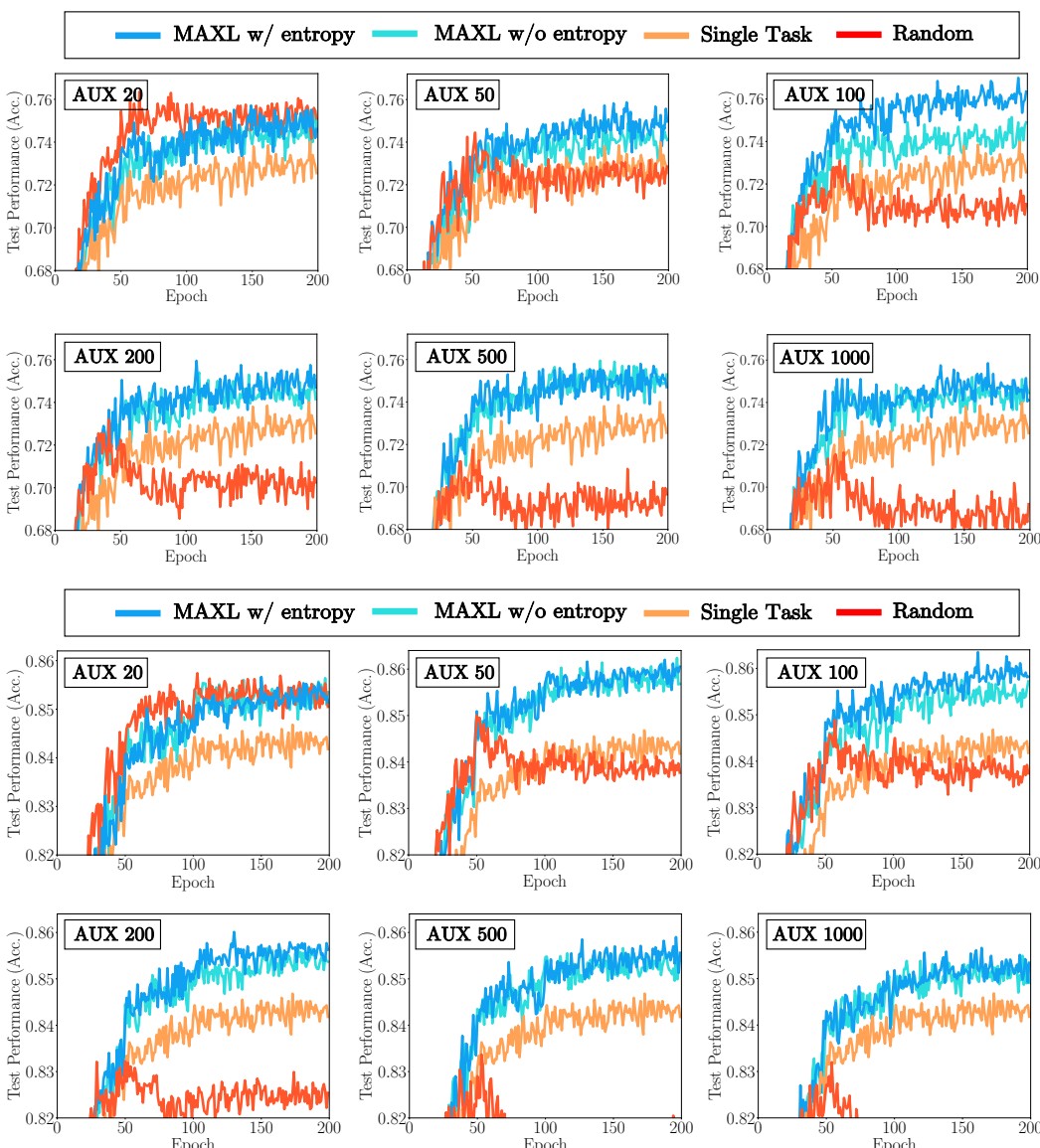

Figure 7: Testing performance on CIFAR10 (bottom) and CIFAR10.1v6 (top) datasets, across 6 different numbers of auxiliary classes.

