# OpenReview forum: "Self-Supervised Generalisation with Meta Auxiliary Learning"
_ICLR.cc/2019/Conference_

### Official Review · AnonReviewer1 · 2018-11-03
**An interesting idea for applying meta-learning to a problem of learning auxiliary tasks in a self-supervised fashion**

**Rating:** 6
**Confidence:** 4

**Review:**

Summary:
The role of auxiliary tasks is to improve the generalization performance of the principal task of interest. So far, hand-crafted auxiliary tasks are generated, tailored for a problem of interest. The current work addresses a meta-learning approach to automatically generate auxiliary tasks suited to the principal task, without human knowledge.  The key components of the method are: (1) meta-generator; (2) multi-task evaluator. These two models are trained using the gradient-based meta-learning technique (for instance, MAML).  The problem of image classification is considered only, while authors claimed the method can be easily applied to other problems as well.

Strengths:
- To my best knowledge, the idea of applying the meta-learning to the automatic generation of auxiliary tasks is novel.
- The paper is well written and easy to read.
- The method nicely blends a few components such as self-supervised learning, meta-learning, auxiliary tasks into a single model to tackle the meta auxiliary learning.

Weakness:
- The performance gain is not substantial in experiments. I would like to suggest to use the state-of-the-arts classifier for the principal task and to evaluate how much gain your method can get with the help of auxiliary tasks. You can refer to the state-of-the-arts performance on CIFAR.
- If the information on the hierarchy of sub-categories is not available, it will be an annoying hyperparameters that should be well tuned.

---

> ### Author Response · Authors · 2018-11-26
> **Response to Reviewer 1**
>
> We thank for the reviewer for their positive comments on our work, and we share our responses below.
>
> The purpose of our work is not to achieve state-of-the-art performance simply by incorporating the latest network architectures and optimisers. Instead, we provide a novel general framework for automating generalisation, and show that when used with standard classification networks across all baselines, our method performs the best.
>
> Furthermore, as we also explained in Reviewer #3, the hyper-parameters for defining a hierarchy is not critical, and we can choose an arbitrary hierarchy whilst still achieving better performance than baselines. In the future work, we would like to explore how to find the optimal hierarchy in an automatic manner, or provide an alternative solution on building a general type of auxiliary tasks (such as regression). However, this is the first work to present a double-gradient method for auxiliary task generation, and we believe that it is important to present the success of this initial method now given how simple and general it is, and then fine-tune other aspects in future work.

---

### Official Review · AnonReviewer3 · 2018-11-03
**Not accurate to call it meta-learning, the auxiliary labels might bring few helpful information, lacking comparisons to several important baselines and benchmark datasets.**

**Rating:** 4
**Confidence:** 4

**Review:**

This paper proposes a self-auxiliary-training method that aims to improve the generalization performance of simple supervised learning. The basic idea is to train the classification network to predict fine-level auxiliary labels in addition to the ground-truth coarse label, where the auxiliary labels used in training is generated by a generator network. During training, the classification network and the generator network are alternatively updated, and the update of the latter aims to maximize the improvement of the former after using the generated auxiliary label for training. The method requires a class hierarchy in advance to define the binary mask applied to the output layer for auxiliary class prediction. A KL divergence term is attached to the optimization objective to avoid generating trivial and collapsing auxiliary classes.

Pros:

1) The main idea is simple and easy to understand.
2) It discusses the class collapsing problem in generating pseudo (auxiliary) labels and provides a reasonable solution, i.e., using KL divergence as regularization.
3) Uses several visualizations to show experimental results.

Cons:

1) The problem it aims to solve is neither multi-task learning nor meta-learning: it tries to solve a supervised classification problem defined on principle classes, with the help of simultaneously predicting/generating auxiliary class labels. Although the concept of "task" is not explicitly defined in this paper, the authors seem to associate each task with a specific class. This is not correct: in meta-learning, each task is a subset of classes drawn from a ground set of classes, and different tasks are independently sampled. In addition, the classification models for different tasks are independent, though their training might be related by a meta-learner. Hence, the claims in multiple places of this paper and the names for the two networks are misleading.

2) At the end of Page 4, the authors show that the update of the generator only depends on the improvement of the classifier after using the auxiliary label for training. In fact, the optimal auxiliary labels minimizing the objective is the ground truth label for principle classes. This results in the class collapsing problem observed by the authors. The KL divergence regularization introduces extra randomness to the auxiliary labels and thus mitigates the problem, but it hardly provides any useful information except randomness. In other words, the auxiliary labels for a specific principle class are very possible to be multiple noisy copies of the principal label with random perturbations. So it is not convincing to me that the auxiliary labels generated by the generator can be really helpful. My conjecture is that the observed improvements are mainly due to the softness of the auxiliary labels, which has been proved by model compression/knowledge distillation and recent "born-again neural networks". To verify this, the authors might need to compare the results with those methods (which use the generated soft probability of ground truth classes for training), and the "random-noisy copies of soft principle label" mentioned above.

3) The experiments lack comparisons to several important baselines from self-supervised learning community, and methods using soft labels for training (as mentioned in 2) above). A successful idea of self-supervised learning is to use the output feature map of the trained classification network to generate auxiliary training signals, since it provides extra information about the learned distance beyond the ground-truth labels. The authors might want to compare to "Mathilde Caron, Piotr Bojanowski, Armand Joulin, and Matthijs Douze. Deep Clustering for Unsupervised Learning of Visual Features. ECCV 2018." and "Carl Doersch and Andrew Zisserman. Multi-task self-supervised visual learning. ICCV 2017." Moreover, since the method is not a meta-learning approach for few-shot learning, it is not fair and also not appropriate to compare with Prototypical Network.

4) Although the paper claims that the ground truth fine labels are not required, it requires a class hierarchy, which in the experiments are provided by the dataset and defined between true coarse and fine classes. In practice, such hierarchy might be much harder to achieve than the primary (coarse) labels, and might be as costly to obtain as the true fine-class labels. This weakens the feasibility of the proposed method.

5) The experiments only test the proposed method on CIFAR100 and CIFAR10, which has at most 100 fine classes. It is necessary to test it on datasets with much more fine classes and much-complicated hierarchy, e.g., ImageNet, MS COCO or their subsets, which have ideal class hierarchy structures.

Minor comments:

Some important equations in the paper should be numbered.

---

> ### Author Response · Authors · 2018-11-26
> **Clarification and other comments**
>
> We thank for the reviewer for their comments on our work, and we share our responses below.
>
> 1) We agree that we did not provide a clear definition of "task". In the present paper there are two tasks: classification into primary labels, and classification into secondary labels. We did not mean to imply that the classification of a specific class is a task on its own. We agree however that a clearer introduction of the terminology would be clearly helpful and we plan to add this to the final submission.
>
> 2) This comment is not entirely correct and we would like to apologies for any confusion in the paper. Actually, the update of the generator depends on the improvement of the classifier for the *principal* labels on the *meta-training* data, i.e. the improvement in generalisation to unseen data. Thus, the optimal auxiliary labels are not the ground-truth labels for the principal classes, since this would make both terms in the minimisation for $\theta_1$ (the second equation in 3.2) identical and not allow any leveraging of the meta-training data. Also, we would argue that the KL-divergence, rather then introducing noise, allows us to avoid collapsing classes which we would claim are due to dying neurons (again, there is not loss/mechanism drawing the auxiliary labels to be the same as the primary ones). These claims are supported by showing that providing random labels does not lead to any improved performance and by our experience that using hard labels does indeed improve performance.
>
> 3) Providing fair comparisons across a range of very different methods is not easy when other methods aim to solve a different problem. Concerning the comparison with prototypical networks, we do agree that this is not a fair comparison and we would like to change the phrasing in the paper. The original reason for associating this to the prototypical network was that we employ their zero-shot setup: i.e. we use a VGG network to obtain prototypes on the meta-data and then use these prototypes to define an auxiliary task on the training-data.
>
> 4) We do agree that requiring the class hierarchy is a current limitation of the work. While it is still general enough for solving classification tasks (we merely have to choose a fixed number of sub-classes per task, e.g. 5 without having to provide anything else), we would want to look at more general auxiliary task in future. One option we are considering is employing an auxiliary regression task, where the generator network would provide vectors and the corresponding loss would be simple regression. However, since this is the first work to use a double gradient method for auxiliary task generation, we believe that presenting results with a comparison to human auxiliary labels, which itself also requires this hierarchy, is a good starting point.
>
> 5) We would very much like to test our approach on more complex datasets with more varied classes, and this will be part of future work. However, we would like to repeat that our approach can work with an arbitrary hierarchy (e.g. assigning the same number of sub-classes to every class). The reason why we only used 100 classes in our experiments is for allowing the comparison with human-defined classes, but in principle we could use any number of sub-classes per primary class. In the CIFAR10 dataset in which a hierarchy is not defined, we show that using 6 different hierarchies all lead to a better generalisation.

---

### Official Review · AnonReviewer2 · 2018-11-06
**The paper is an incremental contribution for an artificially sounding problem**

**Rating:** 4
**Confidence:** 3

**Review:**

This paper proposes an algorithm for auxiliary learning. Given a target prediction task to be learned on training data, the auxiliary learning utilizes external training data to improve learning. The authors focus on a setup where both target and external training data come from the same distribution but differ in class labels, where each class in the target data is a set of finer-grained classes in the auxiliary data. The authors propose a heuristic for learning from both data sets through minimization of a joint loss function. The experimental results show that the proposed methods works well on this particular setup on CIFAR data set.

Strengths:
+ a new auxiliary learning algorithm
+ positive results on CIFAR data set

Weaknesses:
- novelty is low: the proposed algorithm is a heuristic similar to previously proposed algorithms in the transfer learning and auxiliary learning space
- there is no attempt to provide a theoretical insight into the performance of the algorithm
- the problem assumptions are too simplistic and unrealistic (feature distributions of target and auxiliary data are identical), so it is questionable if the proposed algorithm has practical importance
- experiments are performed using a synthetic setup on a single data set, so it remains unclear if the algorithm would be successful in a real life scenario
- the paper is poorly written and sentences are generally very hard to parse. For example, section 3.1 is opened by statements such as "(we use) a multi-task evaluator which trains on the principal and auxiliary tasks, and evaluates the performance of the auxiliary tasks on a meta set"??

---

> ### Author Response · Authors · 2018-11-26
> **Response to Reviewer2**
>
> We thank for the reviewer for their comments on our work, and we share our responses below.
>
> 1. Novelty: To the best of our knowledge, this is the first paper presenting a simple solution to generating useful auxiliary tasks in a self-supervised manner.  The idea indeed was inspired by other works in auxiliary learning, but only to the extent that we also use auxiliary tasks to improve performance of a principal task. The method is not a heuristic; it is theoretically motivated by use of the double gradient, and inspired by the success of this in meta learning (e.g. MAML [1]). If the reviewer thinks our method is an incremental contribution or similar to previous algorithms, please list the specific references.
>
> 2. The theoretical insight in this paper comes from the recent advancements in using a double gradient, such as in MAML [1], or understanding what makes a good auxiliary data sampler [2]. The inner gradient is based on the standard auxiliary learning loss as proposed in other works, whereas the outer gradient uses this inner gradient to actually learn the auxiliary tasks. The use of an outer gradient for auxiliary learning is our key novelty, and has not been used in any works before.
>
> 3. Feature distributions of training and meta-training data (target and auxiliary data in your language) are actually not identical. The "learning to generalise" success from our method is due to closing the *existing* distribution shift in these two datasets. If the distributions are identical, then we wouldn't have any improved generalisation from our method.
>
> 4. Both CIFAR10 and CIFAR100 are the subsets from 80 million tiny images dataset [3]. As described in the website and paper, all images are collected from the internet and partially labelled by humans, and thus indeed present a real-world setup rather than a synthetic setup. Further, we show that if a harder test set with a more variety exists (CIFAR10.1v6), out method could provide even better generalisation (Figure 4). Thus, we hope the reviewer could better explain why you think our algorithm could fail in real-world scenarios.
>
> [1] Finn et al. Model-Agnostic Meta-Learning for Fast Adaptation of Deep Networks ICML, 2017.
> [2] Zhang et al. Fine-Grained Visual Categorization using Meta-Learning Optimization with Sample Selection of Auxiliary Data, ECCV 2018.
> [3] http://people.csail.mit.edu/torralba/tinyimages/

---

### Meta-Review · Area_Chair1 · 2018-12-14

**Confidence:** 4
**Recommendation:** Reject

**Metareview:**

This paper proposes a framework for generating auxiliary tasks as a means to regularize learning. The idea is interesting, and the method is simple. Two of the three reviewers found the paper to be well-written. The experiment include a promising result on the CIFAR dataset. The reviewer's brought up several concerns regarding the description of the method, the generality of the method (e.g. the requirement for class hierarchy), the validity and description of the comparisons, and the lack of experiments on domains with much more complex hierarchies. None of these concerns were not addressed in revisions to the paper. Hence, the paper in it's current state does not meet the bar for publication.